# Application of Red Mud in Wastewater Treatment

**Li Wang [1,2,3], Guangyan Hu [1,2], Fei Lyu [1,2], Tong Yue [1,2], Honghu Tang [1,2,*], Haisheng Han [1,2], Yue Yang [1,2] 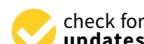, Runqing Liu [1,2] and Wei Sun [1,2,*]**

[1] School of Minerals Processing and Bioengineering, Central South University, Changsha 410083, China; Li-wang@csu.edu.cn (L.W.); hugy0624@csu.edu.cn (G.H.); lvfei2012@csu.edu.cn (F.L.); yuetong@csu.edu.cn (T.Y.); hanhaishengjingji@126.com (H.H.); Eric1911@126.com (Y.Y.); liurunqing@126.com (R.L.)

[2] Key Laboratory of Hunan Province for Clean and Efficient Utilization of Strategic Calcium-Containing, Mineral Resources, Central South University, Changsha 410083, China

[3] State Key Laboratory of Mineral Processing (Beijing), General Research Institute of Mining & Metallurgy, Beijing 100160, China

* Correspondence: honghu.tang@csu.edu.cn (H.T.); sunmenghu@csu.edu.cn (W.S.); Tel.: +86-731-88879622 (H.T.); +86-731-88830482 (W.S.); Fax: +86-731-88710804 (H.T.); +86-731-88660477 (W.S.)

**Abstract:** Red mud (RM) is an industrial waste produced in large amounts during alumina extraction from bauxite. Its disposal generates serious environmental pollution due to high alkalinity. Therefore, a strategy for the effective utilization of RM must be developed. For instance, RM may be transformed into useful products, such as adsorbents. Given its high concentrations of aluminum oxides, iron oxides, titanium oxides, silica oxides, and hydroxides, RM may be developed as a cheap adsorbent for the removal of various ions from aqueous solution and soils (e.g., metal and non-metal ions, phenolic compounds, and dyes) and waste gas purification (sulfide and carbide). This review summarizes the background, properties, and applications of RM as an adsorbent. Proper approaches of removing metal and non-metal elements from wastewater are also systematically reviewed and compared. Emphasis is placed on the surface modification of RM to obtain high adsorption. Finally, the scope for future research in this area for RM is discussed in depth.

**Keywords:** red mud; environmental remediation; polluted water; waste gas; soil

## 1. Introduction

Red mud (RM) is the substantial solid industrial waste produced in alumina production secondary to alkalization and bauxite digestion [1,2]. Approximately 2–3 tons of bauxite is needed to produce 1 ton of alumina. Thus, the amount of RM yield can be estimated by applying the ratio of 1.5 to alumina production data [3]. The global stock of RM was predicted to reach approximately 4 billion tons in 2019, with a production rate of 0.15 billion tons per year [4]. The amount of RM discharged is 0.06 billion tons per year in China, and the stock of RM is over 0.6 billion tons [5]. The annual alumina production data (Figure 1) show that RM production has considerably increased from 2007 to 2018 [6]. In 2018, RM production is approximately 116 million tons worldwide [7]. The grade of bauxite ore continues to decline because of the preliminary exploitation of high-quality bauxite reserves, causing the output of RM to increase. In addition, alumina production has considerably increased worldwide (Figure 1) [8]. China, Oceania, and South America accounted for 54.37%, 15.49%, and 8.27% of the global production of alumina in 2017, respectively, with the rest of the world accounting for approximately 20% (Figure 1) [8].

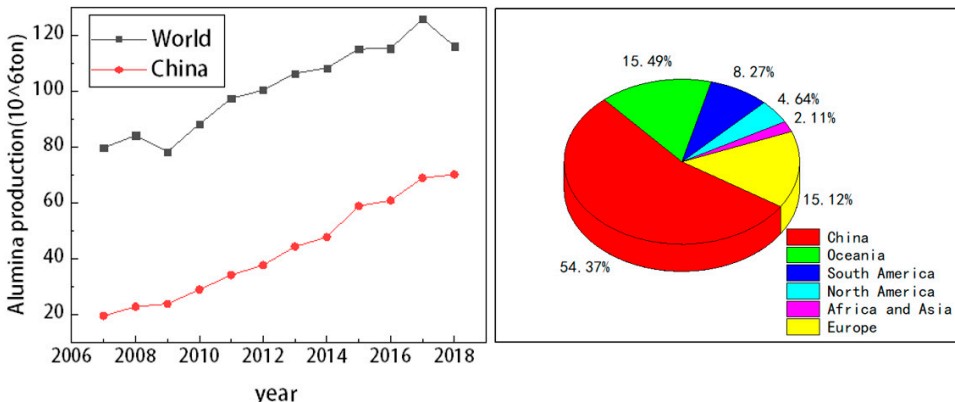

**Figure 1.** Global data from the International Aluminium Institute regarding annual alumina production worldwide and in China. Regional distribution of alumina production in 2017 [5–8]. Reproduced with permission from [5], published by Elsevier B.V., 2016.

RM has not been applied in large quantities until now because of its high alkalinity, technical and economic cost restrictions, industrial conditions, public consideration of its toxicity leaching, and market demand [9]. Moreover, the chemical and mineral components of RM widely vary depending on the origin and processing conditions of bauxite. Therefore, no unified method and standards are available for RM treatment [10]. Thus, RM is mainly stored in dams. Large areas of RM storage occupy considerable land resources and seriously pollute the environment [11].

Methods for the safe treatment and disposal of RM are urgently needed [12]. At present, the treatment and utilization of RM are mainly concentrated on metal recovery, adsorbents, catalysis, building materials, and other aspects [13]. The comprehensive utilization rate of RM is less than 4% in China, indicating that a major proportion of RM is unutilized [14].

RM exhibits a small particle size (particle diameter of 0.075–0.005 mm accounts for about 90%) but large specific surface area (40–70 m$^2$/g) and high chemical reactivity, which benefit the adsorption of metal ions [15]. Moreover, the small amounts of Ca$^{2+}$ and Mg$^{2+}$ in RM can easily precipitate with soluble carbonate in strong alkaline solutions, thereby providing additional adsorption sites for toxic trace elements in adsorption solutions [16]. The relatively strong alkalinity of RM can decrease the mobility of heavy metal ions, which is highly suitable for adsorption to remove toxic and harmful ions in sewage, soil (such as Pb$^{2+}$, Cr$^{3+}$, As$^{3+}$, and F$^-$), and corrosive gases in waste (such as SO$_2$ and NO$_2$) [17]. The harmful effects of RM on the environment can be effectively reduced by transforming RM into absorbent and other useful products [18]. This review summarizes the composition and basic properties of RM and enumerates the applications of RM in environmental restoration, thereby providing a reference for the environmental protection and effective utilization of RM.

## 2. Properties of RM

The composition and properties of RM depend on bauxite ore sources and alumina production processes [19]. The composition and corresponding physical and chemical properties of RM vary depending on the different production methods of alumina and producing areas of raw materials. Table 1 lists the chemical compositions of RM using different production processes (Bayer, sintering, and combined Bayer–sintering process) [19].

The composition of RM is affected by the chemical composition of bauxite and the production of alumina. The main components of RM are Fe$_2$O$_3$, Al$_2$O$_3$, SiO$_2$, CaO, Na$_2$O, and TiO$_2$, which account for approximately 85% of RM [20]. RM produced by the Bayer method has a much higher content of Fe$_2$O$_3$ than that produced by the combined process. RM produced by sintering has a significantly higher content of Al$_2$O$_3$ than that produced by the combined Bayer–sintering process. Meanwhile, RM produced by the combined Bayer–sintering process has a significantly higher content of SiO$_2$ than that produced by the Bayer process. However, the content of SiO$_2$ produced by the combined

Bayer–sintering process was lower than that prepared by the Bayer method. Moreover, decaying components and trace nonferrous metals, such as rhenium, gallium, yttrium, scandium, tantalum, niobium, uranium, thorium, and lanthanide elements, have been observed in RM [21]. The main environmental hazard factor of RM is its $Na_2O$ supernatant. The added solution contains 2–3 g/L alkali and has pH 13–14. The main components are K, Na, Ca, Mg, Al, $OH^-$, $F^-$, $Cl^-$, and $SO_4^{2-}$ [22].

**Table 1.** Chemical compositions of RM using different production methods (%). Reproduced with permission from [23], published by Elsevier Ltd., 2017.

| Chemical Constituent | $Fe_2O_3$ | $Al_2O_3$ | $SiO_2$ | CaO | $Na_2O$ | $TiO_2$ | $K_2O$ | MgO | $Sc_2O_3$ | $Nb_2O_5$ | Loss |
|---|---|---|---|---|---|---|---|---|---|---|---|
| Bayer process | 13.69 | 7.02 | 18.10 | 42.21 | 2.38 | 2.1 | 0.30 | - | - | - | - |
| Combined process | 10.97 | 7.68 | 22.67 | 40.78 | 2.93 | 3.26 | 0.38 | 1.77 | - | - | 11.77 |
| Sintering process | 11.4 | 10.66 | 21.06 | 40.62 | 1.49 | - | 0.45 | 0.93 | - | - | 6.86 |

Table 2 lists the mineral compositions of RM produced using different methods. As shown in the table, the active component of $\beta$-$2CaO \cdot SiO_2$ in the sintering and combined RM accounts for approximately 50% of the total mass. These two kinds of RM can be directly used in the production of building materials. The main mineral groups in the Bayer RM are aluminum sodium silicate, hydrated garnet, calcite, and monohydrated soft bauxite, with high iron content.

**Table 2.** Mineral composition of RM (%, $\omega$). Reproduced with permission from [23], published by Elsevier Ltd., 2017.

| Mineral Composition (Chemical Formula) | Sintering Process | Combined Process | Bayer Process |
|---|---|---|---|
| $\beta$-$2CaO \cdot SiO_2$ | 46 | 43 | - |
| Sodium aluminosilicate $(Na_2O \cdot Al_2O_3 \cdot 1.7SiO_2 \cdot nH_2O) \cdot NaX$ or $Na_2X$ | 4 | 4 | 20 |
| Anorthite $3CaO \cdot Al_2O_3 \cdot 3Si_2O_2$ or $3CaO \cdot Al_2O_3 \cdot xSiO_2 \cdot (6-2x)H_2O$ | 5 | 2 | 20 |
| Calcite $CaCO_3$ | 14 | 14 | 19 |
| Limonite $Fe_2O_3 \cdot H_2O$ | 7 | 4 | 4 |
| Boehmite $Al_2O_3 \cdot H_2O$ | - | 1 | 21 |
| Perovskite $CaO \cdot TiO_2$ | 7 | 12 | 15 |
| $4CaO \cdot Al_2O_3 \cdot Fe_2O_3$ | 8 | 12 | - |
| $Na_2O \cdot Al_2O_3 \cdot 2SiO_2$ | 7 | 8 | - |
| $FeS_2$ | 1 | - | - |
| Others | 1 | - | 1 |
| Total | 100 | 100 | 100 |

Various alumina production processes are used for different bauxite types worldwide due to several mineral compositions and grades (Table 3). The Australian and Indian bauxite are trihydrous and monohydrous soft bauxite, which are produced by the Bayer process. The samples from Shandong, Henan, and Shanxi adopt sintering, Bayer, and combined RM, respectively. The main components, including $Al_2O_3$, $Fe_2O_3$, $SiO_2$, CaO, $Na_2O$, and $TiO_2$ of RM from different sources are basically the same. However, the iron content in RM in China is considerably lower than those in Australia and India. In addition, the contents of $Fe_2O_3$, $Al_2O_3$, and $Na_2O$ in Bayer RM are higher than those in the sintering or combined method. Moreover, the contents of CaO and $SiO_2$ are relatively low. Furthermore, substantial metal oxide keeps the pH of RM between 12 and 14.

**Table 3.** Major chemical components of RM worldwide (mass fraction %). Reproduced with permission from [23], published by Elsevier Ltd., 2017.

| Chemical Constituent | $Fe_2O_3$ | $Al_2O_3$ | $SiO_2$ | CaO | $Na_2O$ | $TiO_2$ | $K_2O$ | MgO | $Sc_2O_3$ | $Nb_2O_5$ | Loss |
|---|---|---|---|---|---|---|---|---|---|---|---|
| Bayer process | 13.69 | 7.02 | 18.10 | 42.21 | 2.38 | 2.1 | 0.30 | - | - | - | - |
| Combined process | 10.97 | 7.68 | 22.67 | 40.78 | 2.93 | 3.26 | 0.38 | 1.77 | - | - | 11.77 |
| Sintering process | 11.4 | 10.66 | 21.06 | 40.62 | 1.49 | - | 0.45 | 0.93 | - | - | 6.86 |

A scanning electron microscope is used to characterize and observe RM [24–26]. RM exhibits strong alkaline composition, complex properties, and abundant metal oxides. Moreover, RM particles show uniform dispersion, thereby indicating that RM has a large specific surface area (40–70 m$^2$/g), porous characteristics (void ratio = 2.53–2.95, average pore size = 2.98–3.82 nm), and excellent stability in aqueous solution [27]. These characteristics render RM suitable as an adsorbent. Applying RM adsorbents in wastewater, waste gas, and soil restoration not only overcomes the negative effect of RM accumulation on the environment but also contributes to environmental restoration.

## 3. Application in Water Treatment

Increased industrial and agricultural activities result in the production of various toxic pollutants, which are the major cause of water pollution worldwide. The types of contaminants in wastewater are mainly determined by the nature of the industry. However, the pollutants commonly found in wastewater are metal and non-metal ions, phenols, and various dyes. The pollutants in wastewater are toxic to aquatic organisms and may cause natural water to be unsuitable for drinking and growing crops. Various methods, such as coagulation, foam flotation, filtration, ion exchange, aerobic and anaerobic treatment, advanced oxidation process, solvent extraction, adsorption, electrolysis, microbial reduction, and activated sludge, have been employed in wastewater treatment [28]. Compared with the more conventional methods, flotation offers several advantages, such as higher metal selectivity, higher removal efficiency, higher over flow rates, lower detention periods, lower operating cost, and production of more concentrated sludge [29]. However, the disadvantages of flotation include high initial capital cost and high maintenance and operation costs Electrochemical technology for heavy metal removal from wastewater is a rapid and well-controlled method that requires few chemicals, provides good reduction yields, and produces less sludge. However, the development of electrochemical technologies is restricted by their high initial capital investment and expensive electricity supply. Membrane filtration can remove heavy metal ions with high efficiency, but its high cost, process complexity, membrane fouling, and low permeate flux limit its use in heavy metal removal [30]. The sludge produced when using coagulation flocculation to remove heavy metals from wastewater features good settling and dewatering characteristics [31]. However, this method involves chemical consumption and increased sludge volume generation. In a word, these technologies demonstrate evident shortcomings, including inadequate removal of pollutants [32], high capital costs [33], considerable reagents or energy requirements [34], and generation of toxic sludge or other wastes requiring further safe disposal [35]. Adsorption is a common process in wastewater treatment technologies. RM, a type of industrial waste with low cost and strong adsorption, is used to treat wastewater discharged by various industries and could provide substantial advantages for environmental pollution control. RM can be used as an adsorbent to remove different pollutants from wastewater, as shown in Figure 2. Prior to water treatment, RM should be first activated by acidification, thermal activation, and synthesis to prevent secondary pollution to water and enhance its adsorption capacity [36].

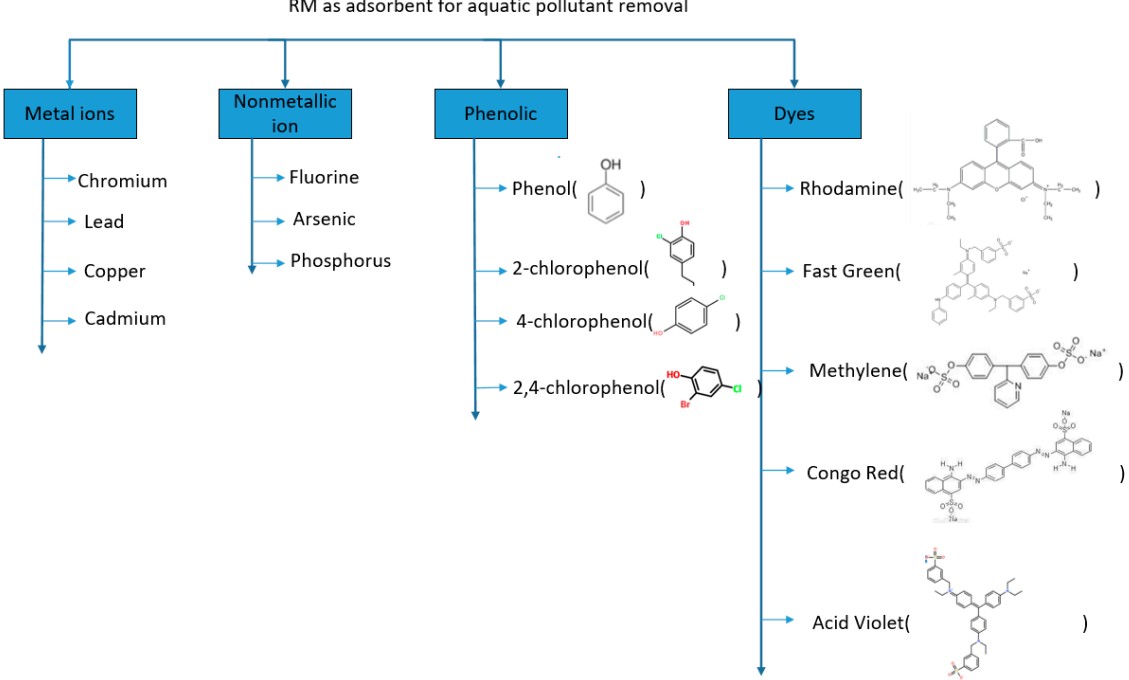

**Figure 2.** RM as adsorbent for the removal of aquatic pollutants from wastewater. Reproduced with permission from [37], published by Taylor & Francis, 2011.

### 3.1. Removal of Metal Ions from Water

Metal ions are aquatic pollutants that pose a serious threat to human health worldwide. The use of RM for metal ion removal from water has been widely explored by various researchers.

### 3.1.1. Removal of Chromium (Cr)

The element Cr should not be ignored because $Cr^{6+}$ compounds are toxic owing to their high water solubility and mobility.

Some researchers explored the use of RM as an adsorbent for the removal of Cr from aqueous solutions [38]. Activated RM (ARM) has been investigated as an adsorbent for $Cr^{6+}$ removal from industrial effluents. ARM was prepared by acid dissolution and then treated by ammonia precipitation [38,39]. The Langmuir monolayer capacity of ARM for $Cr^{6+}$ is 30.74 mmol/g. The adsorption of competing anions on ARM is in the order of $PO_4^{3-} > SO_4^{2-} > NO_3^{-}$. Cr is removed from RM activated by hydrochloric acid, and the removal efficiency of chromate is approximately 70% by the optimal pH value of RM. Kim et al. used RM to remove $Cr^{3+}$ from synthetic wastewater [39] and found that the maximum adsorption efficiency is 99.9% when 1.5 g of RM is used to remove Cr from the solution containing 150 mg $Cr^{3+}$/100 mL.

Li et al. showed that the hydrated garnet ($3CaO \cdot Al_2O_3 \cdot SiO_2 \cdot 4H_2O$) in the original RM decomposes into $Ca_2Al_2SiO_7$ and $Ca_3Al_2O_6$ at high temperatures and that $CaCO_3$ is fully decomposed [40]. These changes in metal phase enhance the activity of RM in water and heighten the effect of $Cr^{6+}$ adsorption [40]. The modification of RM-adsorbed $Cr^{6+}$ is mainly static gravitational pull. RM loses surface, hydration, and bound water in the roasting process. The resistance of RM to adsorb $Cr^{6+}$ is decreased, but the surface roughness of RM particles is improved. In addition, specific surface area is increased to provide substantial active sites.

Sahu et al. studied the adsorption of $Cr^{6+}$ on RM activated by $H_2O_2$ at 500 °C [41]. Results showed that ARM exhibits notable adsorption performance on Cr and could effectively remove $Cr^{6+}$ in water in various concentrations, and the removal rate of $Cr^{6+}$ reaches 87.65%. An experimental study on the adsorption column showed that RM has an industrial application value as an adsorbent. The

adsorption column can be directly treated with $HNO_3$ to desorb the adsorbed metal. The adsorbent can be reused, and the presence of salt in wastewater does not affect the adsorption process.

The use of RM/carbon material to recycle $Cr^{6+}$ from wastewater has been studied [36,38], and chromite ($FeCr_2O_4$) has been produced. The RM/carbon material was prepared by mixing RM pulverized coal by carbon heat treatment, which is a cheap and effective $Cr^{6+}$ removal agent. During this process, $Fe_2O_3$ is decreased to nano-zerovalent iron (nZVI) in RM, and several components, such as $Al_2O_3$ and $SiO_2$, from RM act as dispersants and stabilizers of nZVI. The nZVI materials play a vital role in Cr removal, and the nZVI particles are corroded as observed via transmission electron microscopy to form the core–shell or hollow nanoparticles (Figure 3A). The main component of the core–shell or hollow nanoparticles is iron oxide [42]. The valence state of Cr species of used RM/carbon was analyzed by XPS. In addition, $Cr^{6+}$ was diminished and collected in the form of $Cr^{3+}$ on the surface with the addition of RM/carbon. Scanning electron microscopy (SEM)-EDS showed that Cr–Fe hydroxides are first co-precipitated, and their typical morphological characteristics are shown in Figure 3B [43]. In addition, the pH is increased from 3.0 to 6.8 of the solution with $Cr^{6+}$ removal (Figure 3C). In summary, nZVI is dissolved as $Cr^{6+}$ is decreased to generate $Cr^{3+}$ and $Fe^{6+}$ at a low pH. However, this approach requires substantial $H^+$, thereby resulting in a rapid increase in pH. The increase in solution pH prevents the dissolution of nZVI and the reduction of $Cr^{6+}$. Nevertheless, the collection of Cr–Fe hydroxides precipitated on RM/carbon is promoted to remove $Cr^{6+}$ and dissolve Fe ions from the wastewater. The results show that solution pH greatly affects the RM/carbon removal of $Cr^{6+}$ from wastewater. The initial pH of 3.0 may maintain a balance between the reduction of $Cr^{6+}$ to $Cr^{3+}$ and subsequent precipitation of the generated $Cr^{3+}$ and $Fe^{3+}$ ions (Figure 3C). The removal efficiency of $Cr^{6+}$ with RM/carbon at a constant pH of 3.0 is illustrated in Figure 3D. The removal efficiency of $Cr^{6+}$ decreases from 15.4% to 12.7% within 5 h and remains at approximately 12.2% [44].

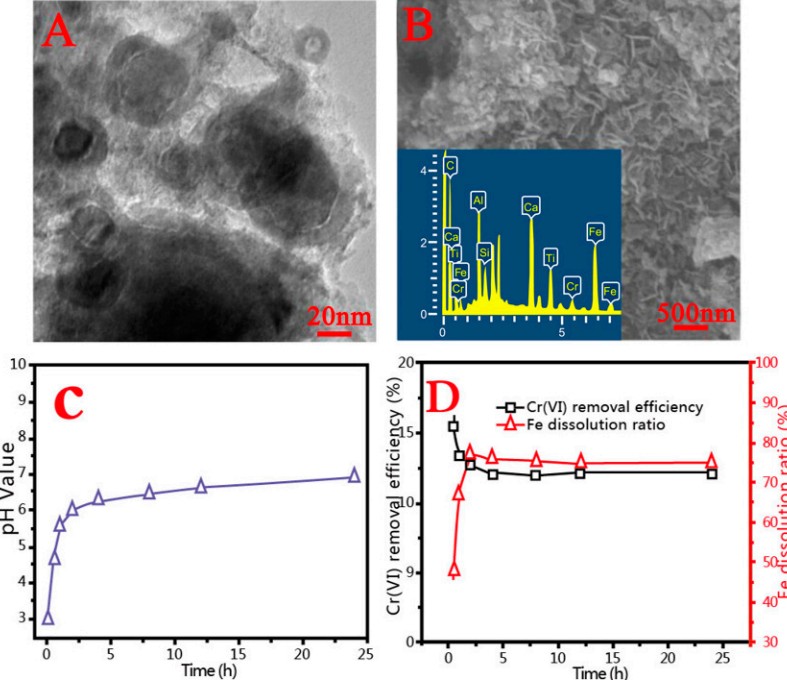

**Figure 3.** TEM (**A**) and SEM-EDS (**B**) images of the spent RM/carbon sample; pH and time during the removal process of $Cr^{6+}$ over RM/carbon at the initial pH of 3.0 (**C**); and removal efficiency of $Cr^{6+}$ and dissolution rate of Fe over RM/carbon at the constant pH of 3.0 (**D**) [40,44]. Reproduced with permission from [40], published by Elsevier B.V., 2018.

The presented material/method exhibits various advantages, such as utilization of RM waste, low material cost, high nZVI activity, and facile carbothermal regeneration of the spent sample. Meanwhile,

$FeCr_2O_4$ with approximately 30 wt. % $Cr_2O_3$ content is regarded as a high-grade mineral resource and exhibits important industrial applications. In conclusion, Cr is recovered and utilized in wastewater, RM solid waste is recycled, and valuable $FeCr_2O_4$ is produced.

### 3.1.2. Removal of Lead (Pb)

The element Pb can hinder the formation of blood cells and accumulate in the human body, resulting in chronic poisoning. Moreover, this element can penetrate the brain tissue through blood and cause brain injury. The state stipulates that the content of Pb in the referenced water should be less than 0.05 mg/L [45].

Yu explored the removal of Pb from aqueous solutions using RM [46]. The effects of different reaction temperatures on 85% adsorbent (the mass ratio of RM, clay, and coal powder is 85:20:5) was investigated to remove $Pb^{2+}$ in simulated wastewater [47]. At the temperatures of 20 °C, 30 °C, and 40 °C and reaction time of 3 h, the removal rates of $Pb^{2+}$ are 60.4%, 82.6%, and 60.4%, respectively, when the initial concentration of Pb is 50 mg/L [46]. Chen et al. studied the effect of RM-adsorbed Pb ions in water [45]. The results showed that RM has excellent adsorption of Pb ions [46]. When the addition amount of RM reaches 2 g/L, the $Pb^{2+}$ removal rate reaches 99.6% [47]. Moreover, temperature exerts a notable effect on adsorption. An increase in temperature causes a faster adsorption rate. This condition indicates that the adsorption reaction of RM on $Pb^{2+}$ is chemical adsorption, and the adsorption process requires an activation energy. Therefore, the adsorption rate can be improved by increasing the temperature [48]. Increasing the temperature can overcome the activation energy of the reaction and promote the adsorption reaction; increasing the temperature also increases the molecular thermal motion in the system and facilitates the adsorption reaction.

### 3.1.3. Removal of Copper (Cu)

The element Cu is found in wastewater from smelting, metal processing, machine manufacturing, organic synthesis, and other industries in China, with the highest amount of Cu discharged from metal processing and electroplating plants, which contain dozens to hundreds of milligrams per liter of Cu [49]. This type of wastewater discharge seriously affects the quality of water body. Cu content of 0.01 mg/L of water can evidently inhibit water self-purification [50], whereas Cu contents exceeding 3.0 mg/L generates a certain odor [51]. If wastewater containing Cu is used to irrigate the farmland, the Pb accumulation of Cu in soil and crops will cause poor crop growth [52]. The critical concentration of Cu in irrigation water is 0.6 mg/L [53]. The high content of Cu ions can contaminate plants and indirectly cause harm to human beings [54].

Yu et al. explored the removal of Cu from aqueous solutions using RM [55]. In the adsorption process, the concentration of $Cu^{2+}$ in solution gradually decreases with prolonged adsorption time. On the contrary, the solution pH gradually increases because of the dissolution of alkaline substances in RM. The interaction between RM and $Cu^{2+}$ is not only adsorption but also precipitation. Moreover, the $Cu^{2+}$ removal rate gradually increases with increasing pH and then reaches equilibrium with the extension of time. The initial adsorption rates at 1.5 and 2 pH are considerably lower than those at pH 2.5 and 3. This result can be ascribed to the high $H^+$ concentration of the RM adsorbent in the solution. Under the low pH condition, the high concentration of $H^+$ in the solution promotes the protonation of functional groups on the adsorbent's surface, the positive downward charge increases on the adsorbent's surface, and the removal rate of $Cu^{2+}$ is decreased in the solution [56].

### 3.1.4. Removal of Cadmium (Cd)

The element Cd is an industrial and environmental pollutant, and its main sources are Zn, Cu, and Pb smelting, electroplating, battery, alloy, paints and plastics, and other industrial productions [57]. Moreover, Cd is a toxic element in the human body. Numerous researchers proposed in the last century 60 s Cd pollution due to Japan's "pain pain disease," and the relationship between environmental Cd pollution and public health has received increasing attention from people.

Duan et al. explored the removal of Cd from aqueous solutions using RM activated by $H_2O_2$ [58]. In the pH range of 2–12, Cd mainly exists in the solution in four forms ($Cd^{2+}$, $Cd^+$, $Cd(OH)_2$, and $Cd(OH)_3^-$) [58]. Therefore, the adsorption forms are different under various pH conditions. In this process, the adsorption mechanism of the RM adsorbent on $Cd^{2+}$ is analyzed as follows:

When the pH is between 2.5 and 6, Cd in the solution mainly exists in the form of $Cd^{2+}$, and its main action mechanism with RM adsorbent is as follows:

$$H^+ + \equiv Fe\text{-}OH = Fe\text{-}OH_2^+, \tag{1}$$

$$Cd^{2+} + \equiv Fe\text{-}OH_2^+ = \equiv Fe\text{-}OCd^+ + 2H^+, \tag{2}$$

$$Cd(OH)^+ + \equiv Fe\text{-}OH_2^+ = Fe\text{-}OCd(OH)^0 + 2H^+. \tag{3}$$

When the pH is between 6.0 and 7.5, the four forms of Cd simultaneously exist.

$$Cd(OH)^+ + \equiv Fe\text{-}OH_2^+ = Fe\text{-}OCd(OH)^0 + 2H^+, \tag{4}$$

$$Cd(OH)_2^0 + \equiv Fe\text{-}OH_2^+ = Fe\text{-}OCd(OH)_2^- + 2H^+, \tag{5}$$

where $Fe\text{-}OH_2^+$ is the protonation site, and $Fe\text{-}OCd(OH)^0$ and $Fe\text{-}OCd(OH)_2^-$ are the compounds formed on the surface of the RM adsorbent. In addition, the surface of the RM adsorbent is negatively charged and exhibits a large specific surface area. Therefore, part of $Cd^{2+}$ is absorbed to the surface of the adsorbent through electrostatic neutralization during adsorption. Some $Cd^{2+}$ is retained inside the adsorbent by pore adsorption. In summary, electrostatic attraction, pore adsorption, and surface chemical reactions are mainly found in the adsorption of the RM adsorbent on $Cd^{2+}$.

Guo et al. studied the use of RM to remove Cd in wastewater by electrostatic attraction [59]. Results show that RM exhibits an excellent adsorption performance to Cd ions, and the adsorption rate reaches 95.32%. The pH value considerably influences the adsorption effect of RM on Cd ions; the higher the pH, the faster the removal rate. This condition is due to the high negative charge and strong acid condition $H^+$ concentrated on the RM surface, but the Cd ions are blocked, thereby resulting in a low removal rate. Wang et al. studied the use of the sintering method of RM particle adsorption agent (GS) to remove Cd [60]. They found that the maximum removal rate of GS on Cd (II) is 100%. Adsorption is mainly dependent on the surface adsorption mechanism and electrostatic attraction of hydroxyl iron, and its adsorption process is in accordance with the pseudo two-stage kinetic model ($R_2 > 0.999$) [61].

At present, RM treated with heavy metals as an adsorbent is mainly in powder form. The powder particles of RM are very small with a specific surface area of 40–70 $m^2$/g. Thus, they can easily absorb other substances, resulting in a high adsorption performance, but are non-conducive to industrial application. The wastewater generated in the activation of powder RM is difficult to treat. In addition, the powder RM is difficult to recover and regenerate after the experiment. Liang et al. proposed a new granulation RM (GRM) adsorbent [62] under controlled experimental conditions at a desired concentration of Cd solution and pH. GRM of about 10 g was added to 100 mL of Cd solution in a conical bottle. The bottles were placed in a vibrator under controlled temperature for 24 h. The adsorption experiments were conducted at 30 °C, 40 °C, and 50 °C. After completion of adsorption, the solution was filtered directly. The adsorption property of GRM to $Cd^{2+}$ in aqueous solution was also studied, the process is shown in Figure 4. On the basis of the pore adsorption and surface chemical reaction of the RM adsorbent to $Cd^{2+}$ experiments, GRM exhibits a strong adsorption capacity for $Cd^{2+}$ and is suitable for industrial applications.

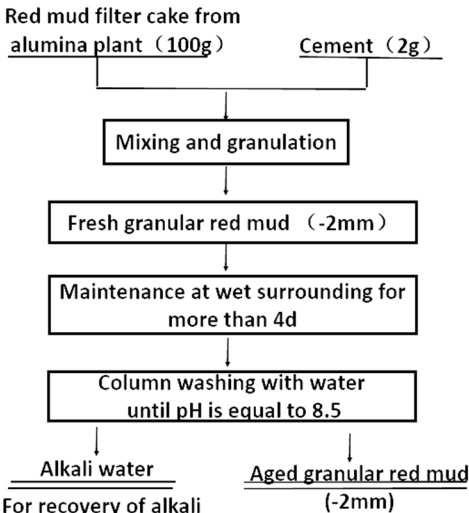

**Figure 4.** Preparation procedure of granular red mud (GRM) adsorbent [63]. Reproduced with permission from [63], published by The Transactions of Nonferrous Metals Society of China, 2012.

## 3.2. Removal of Non-Metallic Ions

### 3.2.1. Removal of Fluorine (F)

The methods for removing excessive fluorinated substances in water include adsorption, ion exchange, precipitation, electrolysis, and electrodialysis. RM can be used as a cheap adsorbent for the removal of fluoride in wastewater.

The ability of ARM activated by HCl to adsorb F is higher than that of raw RM, and the adsorption rate is high (82%) [64]. The adsorption process is consistent with the Langmuir isothermal model. Activation energy can increase the specific surface area and affect the binding sites of RM to improve the adsorption capacity. The adsorption capacity of RM is affected when the pH values are greater than 5.5. Hydroxide and F ions in the solution form intense competition, and the removal rate of F is considerably decreased. The F ion adsorption decreases in acidic environment because of the formation of the weak electrolyte hydrofluoric acid. RM contains certain amounts of CaO, $Al_2O_3$, and $Fe_2O_3$, and these compounds exert adsorption effects on F. Such existence can accelerate the sedimentation rate of flocs, neutralize the alkalinity of RM in the treatment process, and make the effluent pH stable and reach the discharge standard. The HCl and high-temperature calcining activation can remove impurities in the RM channel and surface adsorption and bonded water in the skeleton to dredge the internal pores of RM and decrease the adsorption resistance of water film to ions, which promotes the diffusion and adsorption of RM.

Li et al. used the Bayer RM of Pingguo aluminum factory as the main raw material, which contains certain amounts of $Na_2SiO_3 \cdot 9H_2O$, CaO, and carbon powder, and then prepared RM into balls for roasting [65]. The adsorption capacity is 0.47 mg/g, and the F removal rate reaches 98%. Song-jiang et al. used acid-ARM as the carrier and cerium oxide as the active component [66]. Then, a cerium-loaded adsorbent was prepared using RM. Under 25 °C and static position, the fluoride removal rate can reach more than 98%. Panda et al. studied the effects of modification and adsorption of F ions in wastewater by the Bayer method [67]. Results show that the effect of RM on the removal of F ions in wastewater is evidently improved, thereby making it easy to be recycled.

### 3.2.2. Removal of arsenic (As)

The element As is highly toxic and widely used in industrial and agricultural production. The As pollution causes environmental problems, especially water contamination. Therefore, finding a method to remove As from water with complete, simple, and low-cost operation is necessary. Treating As-containing wastewater and purifying drinking water are of practical importance.

Numerous researchers explored the removal of As from aqueous solutions using RM, which is a great choice. Yan et al. achieved great progress in the removal of As by using solid RM and acid ARM [68]. The removal of As from the liquid phase of RM (LPRM) by coprecipitation with aluminum hydroxide was further studied. The mixture of LPRM–arsenical solution was first neutralized with an acid solution by air-agitation. Then, the mixture was neutralized with $CO_2$ gas. The effect of the volumetric ratio of LPRM:$As^{5+}$-solution on the removal of $As^{5+}$ by coprecipitation as aluminate in the LPRM was studied. Results showed that $As^{5+}$ is effectively removed by LPRM with a volumetric LPRM: $As^{5+}$-solution ratio of 0.1 from an As solution. The removal of As from water by RM under different conditions was also studied. RM efficiently removed $As^{3+}$ in the pH ranges of 7.6–9.0 and 5.5–6.0. With appropriate RM dosage, the residual As in solution is decreased below the regulated acceptable As limit (0.1 mg/L) from industrial wastes.

Activated seawater-neutralized RM (activated Bauxsol, AB) was used to remove inorganic As from water [69]. The AB was prepared by acid and heat treatment. Kinetic data indicated that the $As^{5+}$ and $As^{3+}$ achieve equilibration following pseudo-first-order kinetics. Within the range tested, the optimal pH for $As^{5+}$ adsorption is 4.5, and the efficiency of $As^{5+}$ removal is approximately 100%, irrespective of the initial $As^{5+}$ concentration. Accordingly, the optimum pH for $As^{3+}$ removal is 8.5, and the removal efficiency changes with the initial $As^{3+}$ concentration. The $As^{5+}$ adsorption by AB can be well described using the FITEQL (v.4) and PHREEQC (v.2) computer programs.

RM can be used as an adsorbent of As because it has selective adsorption of arsenate ions, and RM with sorption of arsenate ions can be regenerated by acid or alkali. In industrial production, an integrated treatment system must be studied to remove harmful sediments. Therefore, the system should combine RM adsorption, activated carbon catalyst (oxidation of arsenate ions), and sulfide precipitation methods to form a complete treatment process for removing As from water.

### 3.2.3. Removal of Phosphorous (P)

ARM has an excellent ability of removing P. Li et al. prepared phosphorus-removing adsorbent with RM, which was activated by acid and roasting [70]. The saturated adsorption capacity values of acid- and roasting-ARM on P are 155.2 and 144.2 mg/g, respectively. The adsorption efficiency of acid-ARM increases due to cleaning off the film that was attached to the surface of RM, thereby blocking the adsorption of P and dredging the internal pores of RM. The acidification process causes the erosion of the RM surface and roughness of the external surface. The specific surface area of RM increases from 14.09 $m^2$/g to 21.76 $m^2$/g, thereby increasing the adsorption capacity. The increased adsorption efficiency of RM due to roasting activation leads to the disappearance of moisture, formation of micropores, and increase in specific surface area to 15.69 $m^2$/g.

Li et al. found that the adsorption of P by RM is a monomolecular layer [71]. The authors used ARM particles as the main raw material, fly ash as the activator, bentonite (soap soil) as the binder, and sodium bicarbonate as the foaming agent. The physical and chemical properties of the roasted RM particles were characterized and compared by infrared spectroscopy. Results show that the roasted RM particle surface can be formed with an –OH functional group structure of the material, which can occur with phosphoric acid root in the solution of the ligand exchange reaction and removal of phosphorus. Therefore, RM can effectively adsorb phosphorus in solution.

Zhang et al. used RM as a crystal seed to induce calcium phosphate (HAP) crystallization to recover phosphorus from simulated wastewater [72]. The recovery rate of P can reach 74.1%. They also tried to add RM in the cementitious materials. The effect of RM on the phosphorus purification of ecological concrete was studied. The theoretical maximum phosphate adsorption capacity of RM is 36.76 mg/g, which is greater than those of other adsorbents.

### 3.3. Removal of Phenolic Pollutant

Phenols are regarded as the priority pollutants in organic wastewater. Phenolic compounds are widely used in paper making, pesticides, dyes, textiles, medicine, plastics, rubber, leather making, and

petroleum industries, domestic sewage, and decaying vegetation. Phenol and its derivatives are highly toxic and carcinogenic and must be removed before discharge. RM was used as a potential absorbent for removing phenolic substances from wastewater.

RM was used to remove phenol, 2- and 4-chlorophenol, and 2,4-dichlorophenol from wastewater [73,74]. The maximum adsorption condition of phenol and 2-chlorophenol occurs at pH 6.0, whereas those of 4-chlorophenol and 2,4-dichlorophenol are achieved at pH 5.0 and 4.0, respectively. The adsorption amounts of RM on 2,4-dichlorophenol and 4-chlorophenol are 94.97% and 50.81%, respectively. The researchers also conducted adsorption experiments on the mixtures of phenols and found that the adsorption capacity of 2,4-dichlorophenol is the largest compared with the other phenols. The order of adsorption is 2,4-dichlorophenol > 4-chlorophenol > 2-chlorophenol > phenol, and the adsorption rates achieved are 97%, 93%, 80%, and 51%, respectively. At a flow rate of 0.5 mL/min of column experiments, the removal rate of phenol and its derivatives in RM is up to 98%.

The RM treated with hydrogen peroxide was studied by Gupta [75]. The result is the activated RM of phenol, and the removal rate of its derivatives (2- and 4-chlorophenol and 2,4-dichlorophenol) is 98% at 500 °C. Thus, it can effectively remove phenol from wastewater. The SEM of activated RM can clearly show surface texture and porosity similar to poly-ferric-aluminum silicate. The RM adsorbs phenolic compounds because it has a positive surface charge that attracts negatively charged phenols. A summary of the adsorption capacity of RM for different phenolic pollutants is presented in Table 4.

**Table 4.** Adsorption capacity of RM for the removal of phenolic pollutants from water [37,75–77]. Reproduced with permission from [37], published by Taylor & Francis, 2011.

| Adsorbent | Adsorbate | Amount Adsorbed | Reference |
|---|---|---|---|
| RM | Phenol | 0.63–0.74 mol/g | [77] |
| RM | 2-Chlorophenol | 0.72–0.79 mol/g | [77] |
| RM | 4-Chlorophenol | 0.78–0.82 mol/g | [77] |
| RM | 2,4-Dichlorophenol | 0.80–0.85 mol/g | [77] |
| Neutralized RM | Phenol | $2.50 \times 10^{-5}$ mol/g | [75] |
| Acid-ARM | Phenol | $2.98 \times 10^{-5}$ mol/g | [76] |

### 3.4. Removal of Dyes from Water

Colored dye effluents are generally highly toxic to aquatic biota. Many human health-related problems, such as allergy, dermatitis, skin irritation, cancer, and mutations, are associated with dye pollution in water [78]. Textile industrial dyes are organic pollutants in water. The dyes do not usually undergo photodegradation and oxidative decomposition. Thus, the RM removal of dyes from wastewater is of great importance to water treatment.

Gupta et al. studied the removal of rhodamine B, fast green, and methylene blue dyes from wastewater by using RM [75]. The removal rates of the aforementioned compounds are 92.5, 94.0, and 75.0, respectively, and the optimal pH values for the compounds are 1.0, 7.0, and 8.0, respectively. During the column operation, the removal rate of the compounds is approximately 95–97%, and the flow rate is 0.5 mL/min. The removal rate decreases with increasing flow rate. At the same time, the removal of pugian orange dye at different initial dye concentrations, stirring times, adsorbent dosages, and pH values using RM was studied [75–78]. When the initial pH is increased from 2.0 to 11.0, the removal rate decreases from 82 to 0. The decrease in adsorption amount with pH value is due to the formation of a hydrate complex at the solid solution interface and subsequent acid-base dissociation. At pH 2.0, the maximum removal rate of the dye is 82%.

Ion exchange is the main mechanism of the adsorption process [79,80]. ARM was used to remove the Congo red dye in the batch adsorption experiment. In aqueous solutions, the pH value of the dye solution greatly influences the chemical properties of dye molecules and ARM. When adsorbing ARM, the effective pH is 7.0. The results show that the maximum monolayer adsorption capacity of ARM for Congo red is 7.08 mg/g within 90 min.

Methylene blue was removed by using fly ash and RM as adsorbent. Heat (overnight at 800 °C) and chemical treatment of nitric acid solution were applied to the fly ash and RM samples [81,82]. The adsorption capacity of fly ash and RM is decreased by heating, but the adsorption of fly ash and RM treated by $HNO_3$ is different. Nitric acid treatment improves the adsorption capacity of fly ash and decreases the adsorption capacity of RM. This condition indicates that some organic matter and hydroxyl groups decompose as the temperature increases, thereby decreasing the adsorption capacity. In addition, acid treatment promotes the dissolution of the mineral in carbon, thus increasing the pore volume of the fly ash sample and the adsorption capacity. However, for RM, acid treatment neutralizes hydroxyl ions on the alkaline surface, which is conducive to the adsorption of basic dyes.

A summary of the adsorption capacity of RM for different dyes is presented in Table 5. Evidently, RM exhibits inefficiency for dye removal. Therefore, further research on the removal of different classes of dyes using RM is necessary. Moreover, a substantial understanding of the mechanism of dye adsorption on RM is needed.

**Table 5.** Adsorption capacity of RM for the removal of different dyes from water [37,83–86]. Reproduced with permission from [37], published by Taylor & Francis, 2011.

| Adsorbent | Adsorbate | Amount Adsorbed | Reference |
|-----------|-----------|-----------------|-----------|
| RM | Rhodamine B | $(1.01–1.16) \times 10^{-5}$ mol/g | [83] |
| RM | Fast Green | $(7.25–9.35) \times 10^{-6}$ mol/g | [83] |
| RM | Methylene blue | $(4.35–5.23) \times 10^{-5}$ mol/g | [83] |
| RM | Congo red | $5.81 \times 10^{-6}$ mol/g | [84] |
| RM | Acid violet | $2.42 \times 10^{-6}$ mol/g | [85] |
| Acid-ARM | Congo red | $1.02 \times 10^{-5}$ mol/g | [86] |

## 4. Treatment of Waste Gas

The production of considerable $NO_x$, $SO_x$, and $H_2S$ in various industries leads to serious air pollution. The alkaline substance in RM is abundant, which can purify and adsorb these gases. However, few studies are available on the use of RM in gas purification compared with other applications.

### 4.1. Desulfurization Process

RM can be divided into dry and wet methods for flue gas desulfurization [87]. The phase of RM is conducive to the flue gas desulfurization. The Gibbs free energy of the effective desulfurization component in RM is obtained through thermodynamic analysis of RM desulfurization at room temperature, and the reaction of sulfur dioxide is less than zero. Formulas (6)–(10) show that RM can be used in flue gas desulfurization, where r = reaction, m = mol.

$$Na_2O + SO_2 \rightarrow Na_2SO_3 \qquad \Delta rGm \ (298 \ K) = -333.138 \ kJ/mol \tag{6}$$

$$4Na_2O + 4SO_2 \rightarrow 3Na_2SO_4 + Na_2S \qquad \Delta rGm \ (298 \ K) = -1447.343 \ kJ/mol \tag{7}$$

$$4CaO + 4SO_2 \rightarrow 3CaSO_4 + CaS \qquad \Delta rGm \ (298 \ K) = -818.796 \ kJ/mol \tag{8}$$

$$Al_2O_3 + 4.5SO_2 \rightarrow Al_2(SO_4)_3 + 1.5S \qquad \Delta rGm \ (298 \ K) = -166.869 \ kJ/mol \tag{9}$$

$$2Fe_2O_3 + 5SO_2 \rightarrow 4FeSO_4 + S \qquad \Delta rGm \ (298 \ K) = -314.49 \ kJ/mol \tag{10}$$

4.1.1. Dry Desulfurization

The dry method for flue gas desulfurization uses some components to adsorb pollutants in RM.

RM is used for sulfur dioxide absorption and purification. Scientists found that 1 kg of RM can absorb 11.3 g $SO_2$ at dry desulfurization and that its desulfurization rate is approximately 50%. In wet desulfurization, the $SO_2$ adsorbed by the same RM is 16.3 g, and the desulfurization rate is approximately 90% [88].

In China, Liu et al. used RM adsorption fluid to absorb and purify $SO_2$ flue gas, and the $SO_2$ absorption efficiency is higher than 98% [89]. Zhou et al. used Bayer RM to absorb $SO_2$ in a bubbling reactor [90]. Results show that RM can effectively absorb $SO_2$ at a highly close liquid/gas ratio. Therefore, RM can be used to desulfurize industrial waste gas and decrease environmental pollution. Ghosh et al. used RM to synthesize a compound RM desulfurizer with extremely high desulfurization activity through experimental studies [91]. The amount of RM is greater than 70%, the desulfurization efficiency is over 90%, and the sulfur capacity is over 15%. The pore size of the RM desulfurizer is within the range of 0.94–4.3 nm, and the effect of micro-pore desulfurization is the best. The more pores in the pore size range, the larger the sulfur capacity of the RM desulfurizer. The micropores on the surface of the desulfurizer particles are evenly distributed, and these pores can effectively transport $SO_2$ to the interior of the particles, thereby providing a sufficient reaction surface and creating favorable conditions for optimal matching of diffusion and chemical reaction.

### 4.1.2. Wet Desulfurization

RM was used in the wet process for flue gas desulfurization, and its adsorption fluid reacts with acid gas. RM contains considerable alkaline substances, such as $Al(OH)_3$ and $CaCO_3$, which can react with an acidic gas. Most particles are amorphous and agglomerative, which form an agglomerative space. Therefore, RM has considerable pores, large specific surface area, and excellent adsorption capacity. In the RM adsorption of sulfuric acid fog, acid–base neutralization, surface adsorption, and pore filling may also play notable roles.

The use of sintering RM and activated carbon as raw materials to prepare ARM desulfurizer (ARMD) by thermal activation was studied to improve the desulfurization effect [92]. In the preparation process of ARMD, the effects of the ratio of activated carbon and RM (C/R), temperature, and activation time on the desulfurization efficiency were investigated. In the test, the authors used a 300 mm × 80 mm plexiglass bubbling absorber with an effective height of 200 mm. The structure of the test device is shown in Figure 5. The experimental gas and desulfurization slurry were prepared by $SO_2$ standard gas and air and ARMD and deionized water in proportion, respectively. The test gas was injected into the buffer bottle by the $SO_2$ gas tank and air pump with a constant flow and then evenly mixed. The mixed gas enters the bubbling absorption at the bottom of the tower and reacts with the desulfurization slurry, which is then discharged after being absorbed by the exhaust gas absorption device. The results show that the best preparation conditions of ARMD are as follows: C/R = 1:20, 900 °C thermal activation temperature, and 15 min thermal activation time. When the pH value is 4.5, RM desulfurization efficiency can reach 86.9%.

Leal et al. studied the adsorption effect of RM on sulfuric acid fog by using fixed-bed dynamic adsorption [93]. The removal rate of RM on sulfuric acid fog is high (more than 95%) [92,93]. Qu et al. conducted desulfurization experiments by using sintered RM slurry [94]. XRD analysis results of RM desulfurization reaction before and after the experiments showed that the portlandite, calcite, and hydrated garnet phases completely disappear, but considerable calcium sulfite, gypsum, and other new phases appear. This result indicates that during the chemical reactivity, the reaction compositions of the solid phase in the desulfurization reaction are mainly $Na_2O$, CaO, and $Al_2O_3$. The reaction ends when the residual $Na_2O$ mass fraction is decreased to less than 1%.

Buda et al. studied the flue gas desulfurization of RM [95,96]. The authors conducted an orthogonal experimental study on the liquid–solid and gas–liquid ratios, gas flow rate, and gas temperature influence on the desulfurization efficiency. The results show the influence of various factors on the desulfurization efficiency of size for primary and secondary gas–liquid ratio > solid–liquid ratio > flue gas temperature, and the excellent process conditions are 7:1 liquid–solid ratio, 3.6 $m^3$/h flue gas rate, 15:1 natural gas liquid proportion, and 20 °C flue gas temperature. The research results and device design provide theoretical basis and technical support for the industrialization promotion of RM flue gas desulfurization.

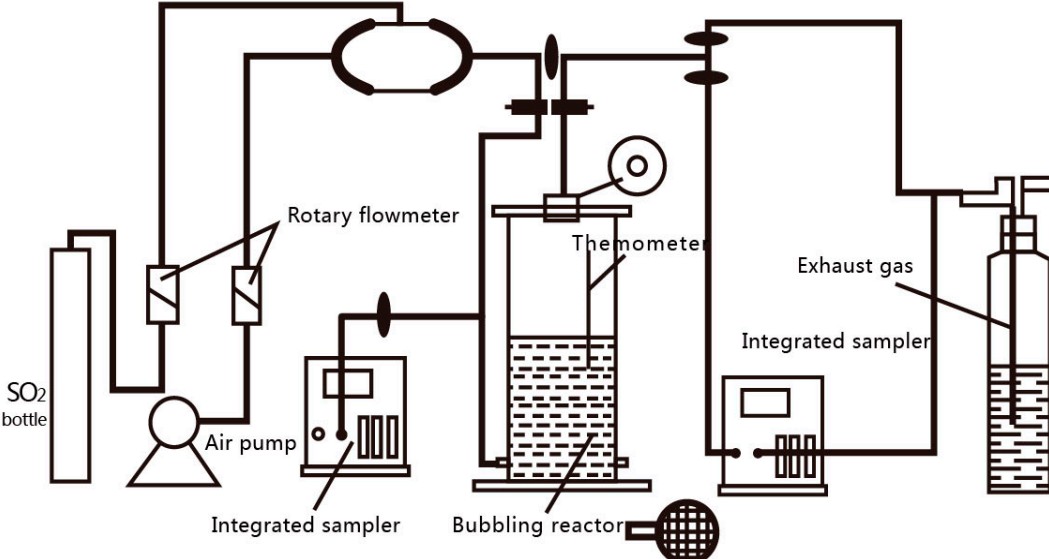

**Figure 5.** Structure of the test device.

Fois et al. conducted an engineering test on Bayer RM desulfurization by using an absorption system composed of supply, $SO_2$ absorption, flue gas, and auxiliary treatment schemes [97]. Red slurry, an absorbent, was mixed with flue gas in the absorption tower. The $SO_2$ in flue gas was chemically reacted with red slurry, and air was pumped into it. The final reaction product is RM containing sulfate to fix sulfur using RM. A patent of "a way to use RM disposal method of flue gas and recycling metal iron, aluminum," was proposed by Lu [98]. First, flue gas was mixed with RM to complete the dealkalization of the RM separation of mud filtrate. Then, the flue gas was mixed with supernatant on RM filtrate, which is obtained by pH control. Thus, iron and aluminum hydroxide should precipitate to realize the separation of iron and aluminum and obtain their recovery rates of 70% and 72% or higher, respectively.

### 4.2. Decarburizing Process

Alharthi et al. used original and neutral RM to absorb $CO_2$ [99]. After carbonization, the total alkalinity of the aforementioned RM is decreased by 85% and 89%. The authors also used the activated RM after $CO_2$ carbide to perform zinc (II) adsorption experiment. The results exhibit a zinc absorption (II) capacity of 14.92 mg/g after high temperature roasting activation. Wang et al. used Bayer RM to capture $CO_2$ [100]. Under the best experimental conditions, the alkali removal rate of RM is up to 42% [101]. The aim of waste gas treatment can be achieved by using RM from solid waste to absorb $CO_2$ from industrial waste gas. Babuponnusami et al. studied the solidification and storage capacity of RM for $CO_2$ under different pressures and the chemical transformation behavior of mineral phases during $CO_2$ solidification [100,101]. The mineral phases formed after RM solidified $CO_2$ are mainly calcium nephrite and calcite ($CaCO_3$). Thus, RM exhibits an excellent effect as $CO_2$ curing agent.

Chaerun et al. [102] utilized RM as an alkaline absorbent for the treatment of sewage sludge in a green supercritical water oxidation system. They concluded that the green disposal method significantly reduces the $CO_2$, $SO_2$, and nitrogen oxide levels in the supercritical water oxidation disposal by 800, 15, and 12 ppm, respectively. The process to treat sewage sludge using RM is shown in Figure 6.

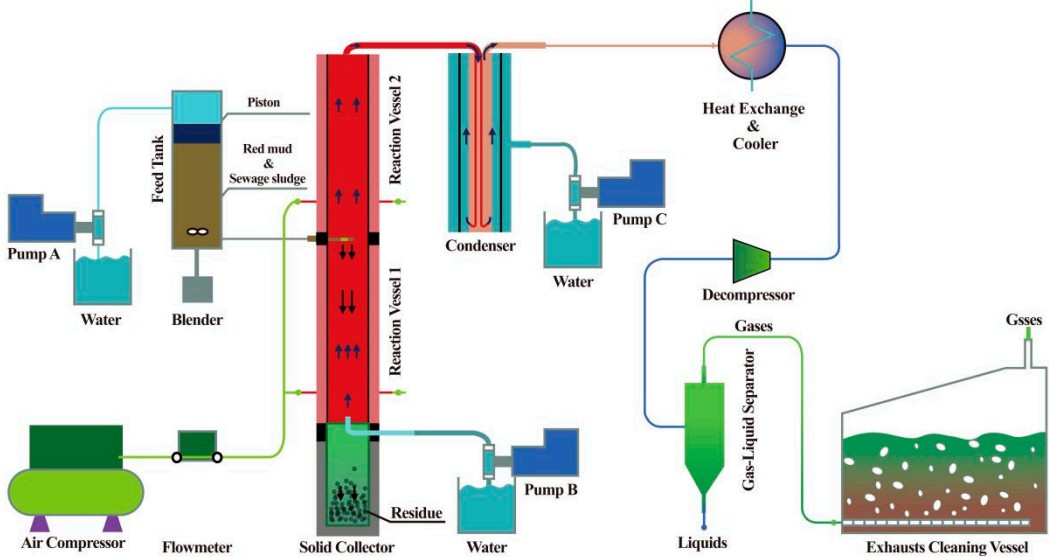

**Figure 6.** Supercritical water oxidation system for sewage sludge treatment using red mud. Reproduced with permission from [81], published by Elsevier Ltd., 2016.

## 5. Remediation of Heavy Metal Contaminated Soils

At present, the deterioration of regional agricultural environment and heavy metal pollution of agricultural products is highly serious, especially in developed areas. Statistics show that Cd, As, Pb, Cr, Hg, and other heavy metals contaminate farmland approximately 20 million hm$^2$, thereby accounting for approximately 1/5 of the total arable land [103]. Therefore, passivation removal of heavy metals in soil is of great importance.

Many studies using RM to remediate metal-contaminated soils have shown that decreasing metal concentrations in plants directly or indirectly consumed by human beings through food crops is beneficial to human health [104]. The metal concentration of plants in potted and field compared with those grown in unrepaired soil decreases by more than 50%.

RM can effectively decrease the amounts of Cd, Pb, and Zn in plants grown in metal/metalloid contaminated soils. The metal concentration in plant tissues (rice grains) decreases from 0.25% to 1.25% with increasing RM application [105]. According to the comparison result of the contents of Cd, Pb, and Zn in pea (*Pisum sativum* L.) and wheat (*Triticum vulgare* L.), the metal content in plants grown with RM is considerably decreased [106]. Also, the addition of RM increases the proportion of metal accumulation in the roots compared with the buds of the two plants. The translocation factors (TFs, metal concentration in shoot/root) for Cd, Pb, and Zn in peas grown in RM-amended soil are 0.086, 0.028, and 0.279, respectively, which are considerably lower than those in peas grown in unamended soil (0.384, 0.052, and 0.825, respectively). Similar changes in metal TFs after RM modification were also reported in wheat plants.

RM considerably decreases the absorption and accumulation of As in the ground and root of maize. Moreover, RM and compost can fix Zn in soil and decrease the element's plant effectiveness. RM and compost were applied to contaminated soils, and the effect of Zn on fixed soils is better than that of separately adding compost.

The effects of RM and sludge on the chemical morphology of Zn in soils were studied by Huang et al. [107]. Results showed that the use of RM alone can effectively decrease Zn content in the soil, and the effect of using RM or sludge to stabilize Zn is better than that of RM and sludge, which cannot only promote the effective state Zn conversion to the non-available Zn but also increase soil fertility. Huang et al. studied the effects of applying RM and lime on maize uptake and accumulation of Cd [108] and found that RM and lime considerably improve the soil pH value and decrease the Cd content in the soil and the absorption and accumulation of Cd. The effects of soil RM and pig manure matching on

the rice antioxidant enzyme system and Cd uptake in paddy soils were studied by field experiments. Results showed that rice yield increases in a certain amount of RM/pig manure.

RM can also decrease metal production in vegetables [109]. The concentrations of As, Cd, Pb, and Zn in lettuce (*Lactuca sativa* L.) grown in soil decrease by 32.8%, 83.5%, 35.4%, and 81.0%, respectively, compared with unamended soil. The effects of sludge and RM on soil physicochemical properties, rape growth, quality, and nutrient status were studied in systems. Data showed that the application of sludge and RM can improve the growth status, yield, and absorption of nitrogen, phosphorus, and potassium and the nutritional status of rapeseed [110]. Improving soil properties is advantageous for the improvement of soil organic matter, nitrogen, and phosphorus nutrient contents to regulate soil pH and conductivity.

The use of RM has another positive effect on contaminated soils [111]. This approach could decrease the vast volumes of RM in storage. Therefore, using RM as a soil amendment is an effective way to decrease RM storage. In summary, a decision tree scheme was proposed to evaluate the applicability of RM as an adsorbent for the treatment of contaminated soil (Figure 7).

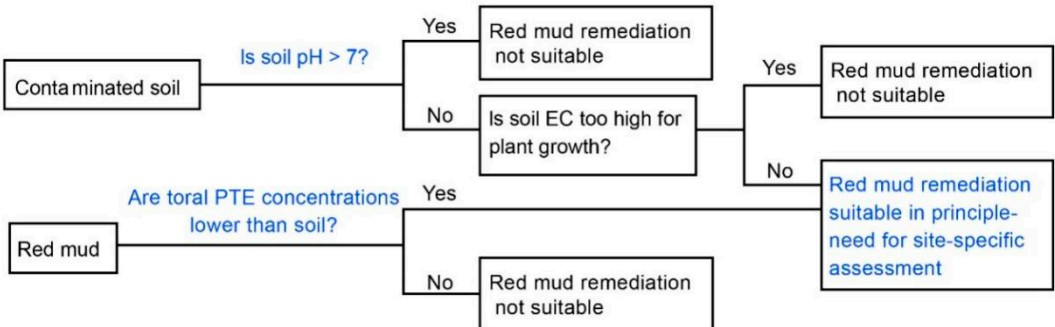

**Figure 7.** Proposed scheme for assessing the use of RM to treat the contaminated soil. Reproduced with permission from [5], published by Elsevier B.V., 2016.

Although RM has made considerable progress in repairing heavy metal soils, consideration of its use in food or forage crops could bring contamination into the human food chain, and further research is needed. Amendments could be used to grow plants that are unsuitable for human consumption in contaminated soil, and RM improvement of contaminated soil would have broader benefits for ecological restoration.

## 6. Conclusions

RM is widely used as an adsorbent in the effective removal of heavy metal and non-metal ions, organic pollutants, and dyes in wastewater treatment as well as sulfides and carbides in waste gas treatment, in the adsorption of pollutants in soil, and in the restoration of contaminated soil. This approach achieved excellent results. However, direct use of raw RM as an adsorbent has limited adsorption capacity. Pretreatment methods, such as acid activation, alkali activation, and heat treatment, are mainly adopted to improve its adsorption capacity. In the future, the manner of finding cheap and efficient modification methods is an important research direction. Moreover, the mechanism by which RM treats various pollutants in water as an adsorption material needs to be elucidated. RM contains rich valuable metals and active ingredients, such as iron, aluminum, and silicon. Maximizing these characteristics to prepare new high-efficiency adsorbents and improve the added value of products is also an important research topic. Main problems existing in the comprehensive utilization of RM are as follows:

1. Methods of recycling or disposal after adsorption of heavy metals to prevent secondary pollution.
2. Study of adsorbents that are more conducive to storage and transportation.
3. Further improvement of adsorption performance (such as making porous nanostructures).

4.    Strategies to avoid secondary pollution of RM (leaching of heavy metals) in adsorption.

**Author Contributions:** Conceptualization, W.S. and H.T.; methodology, L.W.; software, G.H.; validation, L.W., R.L. and F.L.; formal analysis, H.H.; investigation, Y.Y.; resources, H.H.; data curation, Y.Y. and T.Y.; writing—original draft preparation, F.L.; writing—review and editing, G.H.; visualization, L.W.; supervision, H.T. and T.Y.; project administration, W.S; funding acquisition, H.T.

**Funding:** This research was funded by National Key Scientific Research Project (2018YFC1901901); the Natural Science Foundation of China (NSFC, 51704329); the project of Sublimation Scholar's Distinguished Professor of Central South University; the National 111 Project (No. B14034); Collaborative Innovation Center for Clean and Efficient Utilization of Strategic Metal Mineral Resources; Key Laboratory of Hunan Province for Clean and Efficient Utilization of Strategic Calcium-containing Mineral Resources (No. 2018TP1002); Open Foundation of State Key Laboratory of Mineral Processing (BGRIMM-KJSKL-2019-18).

**Conflicts of Interest:** The authors declare no conflict of interest. The funders had no role in the design of the study; in the collection, analyses, or interpretation of data; in the writing of the manuscript; or in the decision to publish the result.

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
