# Peer review of "Application of Red Mud in Wastewater Treatment"

_minerals, doi:10.3390/min9050281_

Round 1
Reviewer 1 Report
1. Line 46, 49. You write … yard … Why? Maybe better to write residue storage or just storage?
2. Fig.1. Why values only until 2016? It is necessary to include 2017 and 2018 years.
3. Line 57. What value of particle size and specific area?
4. Table 1. Why in RM so big CaO content and so low Fe2O3?
5. Table 2. Total amount of phase must be 100%, not 96%
6. Line 119-122. … evident shortcomings, including inadequate removal of pollutants, high capital costs, considerable reagents or energy requirements, and generation of toxic sludge or other wastes … Its necessary to add value of this information.
7. Fig.2. You must add formulas of Phenolic and Dyes.
8. Line 156. What value of Cr6+ efficiency removal?
9. Line 195-197, 213-217. You need to add references of this information.
10. Line 270-271. What’s the value of surface area? “large” is not correct.
11. Table 4, Table 5. Amount adsorbed write in mol/g and mg/g. You must select a single way of writing.
12. Line 434. ∆rGm. What does the r and m mean?
Author Response
I have provided a point-by-point response1 to the reviewer’s comments.

Reviewer 2 Report
The manuscript is a review on the known and possible applications of red mud. The review is well organized, broad and a lot of interesting and relevant references are included. The english language is at a good level, however it needs to be improved in some places and some typos can be removed.
Some sections can be improved and thus my remarks are given below:
I am not really convinced that this review is suitable for Minerals Journal as it deals less with mineral resources. It rather concentrates on the adsorption properties of RM. What is more the title is not fully informative: Environmental Remediation is a too general expression. The authors should propose
a more precise title.
Line 49: Figure 1 - the legend of the figure needs to be corrected: use "North America" instead of "In North America" and "Europe" instead of "European".
Line 89: Please explain TREO abbreviation in table 1. It can be not obvious for all readers in the future.
Line 93-94: The word "adopt" is wrongly used here. Please rearrange the sentences.
Line 101-102 - It would be important to add textural parameters of RM which are reported in literature e.g. specific surface area, information on the porosity (pore volume and pore size distribution). This is particularly important when discussing their use as adsorbents.
Line 134: Cr instead od CR.
Line 349 - I don't think that RM, bentonite and fly ash can be considered as factors. Please make the sentence more clear.
Line 443: In my opinion more information on the microporosity in this case should be provided in the review with comparison with other materials and nanomaterials if possible.
Line 478: This is an example of citation where the year of publication is missing. Please update this and other similar references throughout the manuscript.
Line 583: The second point in conclusion is not clear enough. Please rearrange the sentence.
With regards,
Reviewer
Author Response
I have provided a point-by-point response to the reviewer’s comments.
